# Graphene-Based Materials Prove to Be a Promising Candidate for Nerve Regeneration Following Peripheral Nerve Injury

**DOI:** 10.3390/biomedicines10010073

**Published:** 2021-12-30

**Authors:** Mina Aleemardani, Pariya Zare, Amelia Seifalian, Zohreh Bagher, Alexander M. Seifalian

**Affiliations:** 1Biomaterials and Tissue Engineering Group, Department of Materials Science and Engineering, Kroto Research Institute, The University of Sheffield, Sheffield S3 7HQ, UK; maleemardani1@sheffield.ac.uk; 2Department of Chemical Engineering, University of Tehran, Tehran 1417935840, Iran; pariyazare@ut.ac.ir; 3Department of Surgery and Cancer, Imperial College London, London W12 0NN, UK; a.seifalian@ic.ac.uk; 4ENT and Head and Neck Research Centre, Hazrat Rasoul Akram Hospital, The Five Senses Health Institute, Iran University of Medical Sciences, Tehran 16844, Iran; 5Nanotechnology and Regenerative Medicine Commercialization Centre (NanoRegMed Ltd.), London BioScience Innovation Centre, London NW1 0NH, UK

**Keywords:** graphene-based materials, nervous system, nerve tissue engineering, nerve proliferation, nerve differentiation, surgery, plastic surgery, regenerative medicine, biomedicine, functionalized graphene oxide, drug delivery, spinal cord injury

## Abstract

Peripheral nerve injury is a common medical condition that has a great impact on patient quality of life. Currently, surgical management is considered to be a gold standard first-line treatment; however, is often not successful and requires further surgical procedures. Commercially available FDA- and CE-approved decellularized nerve conduits offer considerable benefits to patients suffering from a completely transected nerve but they fail to support neural regeneration in gaps > 30 mm. To address this unmet clinical need, current research is focused on biomaterial-based therapies to regenerate dysfunctional neural tissues, specifically damaged peripheral nerve, and spinal cord. Recently, attention has been paid to the capability of graphene-based materials (GBMs) to develop bifunctional scaffolds for promoting nerve regeneration, often via supporting enhanced neural differentiation. The unique features of GBMs have been applied to fabricate an electroactive conductive surface in order to direct stem cells and improve neural proliferation and differentiation. The use of GBMs for nerve tissue engineering (NTE) is considered an emerging technology bringing hope to peripheral nerve injury repair, with some products already in preclinical stages. This review assesses the last six years of research in the field of GBMs application in NTE, focusing on the fabrication and effects of GBMs for neurogenesis in various scaffold forms, including electrospun fibres, films, hydrogels, foams, 3D printing, and bioprinting.

## 1. Introduction

Millions of people are suffering from neurodegenerative disorders or acute injuries of the nervous system globally. The nervous system is divided into the central nervous system (CNS) and the peripheral nervous system (PNS). The CNS consists of the spinal cord and brain, and the PNS consists of all of the nerves and ganglia outside of the CNS (Figure 1) [1,2]. The PNS is statistically more susceptible to injury, compared to the CNS, secondary to disease and trauma, leading to neurite damage and neuron loss [3]. Each year, it is estimated that 18 per 100,000 people suffer from peripheral nerve injury (PNI) in developed countries, with the rate thought to be greater in developing countries [4,5]. PNI can permanently impact nervous functions, such as mobility and sensation, as mature neurons are terminally differentiated with no further cell division [6]. Spontaneous axonal regeneration in PNI has been seen in small gaps; however, regenerated nerve function is restricted, particularly in long-gap injuries. Following injury, the axonal membrane breaks apart, causing degradation of the myelin sheath and infiltration of macrophages; within the site of degeneration, the macrophages clear the debris where axonal regeneration should start. Schwann cells are the main glial cells in the nervous system and the PNS, providing a pathway for axonal regeneration and remyelination [7]. Schwann cells are available in PNS and play a critical role when an injury occurs, responding to trauma by detaching the myelin sheath and phagocytosing debris, encouraging cellular proliferation, and releasing further cytokines to continue to recruit inflammatory cells to the damaged area [8,9]. The mature myelinating Schwann cells dedifferentiate by regaining the expression pattern related to immature Schwann cells, proliferate, and then redifferentiate, which induces nerve repair (Figure 2) [10]. On the contrary, due to the lack of Schwann cells in the CNS, the clinical treatments for CNS injuries are ineffective [1].

There are different types of nerve injury with various factors involved, creating a challenge against ‘one size fits all’ conventional treatment and promoting precision treatment tailored for the injury suffered by each patient. Several strategies offer potential treatment for neural disorders, including anti-inflammatory (M2) medications, physiotherapy, nerve grafting, and rehabilitation. Among these methods, autologous superficial cutaneous nerves are known as the gold standard for bridging the nerve gap (>30 mm); however, there are several drawbacks, including major surgical risks, autologous graft rejection, infection, donor shortage, and the likelihood of further surgeries [11]. Tissue engineering (TE) proposes a promising alternative approach to overcome the existing challenges in tissue regeneration [12,13,14]. To successfully achieve neural regeneration, various factors, including topographical, chemical, mechanical, and electrical cues, should be considered. There have been ongoing attempts to develop biomaterial-based therapies to regenerate dysfunctional neural tissues, primarily for a damaged PNS and spinal cord [15,16].

Graphene (G) has a closely packed honeycomb lattice consisting of a sheet of sp^2^-bonded carbon atoms with a high surface area (almost 2600 m^2^/g). The main advantage of graphene is currently the thinnest, strongest, and lightest material known. It is a single layer carbon atom, which is highly thermal and electrically conductive. The disadvantage of graphene as a catalyst is its susceptibility to oxidative environments and biological environments due to features such as jagged edges that can easily pierce cell membranes. The latter property has advantages as well, for its antibacterial activity. The G has been recognized as an attractive candidate to repair injured nerves (Figure 3) [17,18]. G can be synthesized based on top-down and bottom-up approaches. Despite the advantages of G between all other carbon-based materials, it has some limitations, such as unstable chemical structure and insufficient active sites, which limit interaction with other biomolecules. To overcome these restrictions, the chemical modification of G is highly recommended. Graphene oxide (GO) is the primary G derivative that possesses a higher ability to absorb biomolecules due to the presence of carboxyl groups (-OOH) on the edge of its structure, as well as epoxy (-O) and hydroxyl (-OH) groups on the basal plane (Figure 3). GO is synthesized by both hummer and modified hummer methods. Reduced graphene oxide (rGO), another significant G family, is produced by reducing the amount of oxygen in GO through thermal, chemical, or UV exposure processes [15]. rGO is hydrophobic, and it is used as a nanofiller or for coating medical devices. The most interesting and useful derivative of G is functionalized GO (FGO). Usually, it is functionalized by the amine group. FGO can be covalently bonded to the polymer structure, make the graphene-based materials (GBMs) stronger and more uniform; and inherit all the superior properties of graphene, such as conductivity and strength. The disadvantage of FGO is it is more expensive to produce for industrial applications, but the price is reasonable for biomedical applications.

GBMs can enter the body with different routes and penetrate through physiological barriers like the blood–air barrier, blood–testis barrier, blood–brain barrier (BBB), and blood–placenta barrier; after that, they can locate in cells, tissues, and organs, and finally are excreted. This entrance can result in toxicity and genotoxicity, and various factors affect it, including the lateral size, surface structure, functionalization, charge, impurities, and aggregations. Additionally, among mentioned different parameters, the interaction between nano-size GBMs, such as nanoparticles (NPs), nanoflakes and nanosheets, and biological samples (in vitro, in vivo, and clinical trials) is the most crucial parameter for toxicity. Generally, it has been represented that smaller nano-size GBMs induced greater toxicity levels [19,20]. Among several mechanisms underlying GBMs toxicity, the production of reactive oxygen species (ROS) within cells can lead to interactions with biomolecules (e.g., DNA and RNA) [15,21]. However, GBMs have flexible structures that can be modified with other substances, including polymers and biomolecules, to enhance their biocompatibility. GBMs composites have represented an appropriate interaction with DNA and RNA for sensing and drug delivery approaches [2,22]. The oxygen-based functional groups, GO and rGO, have a lower tendency to form aggregates, even after functionalization, which allows them to cross BBB and improves their stability in blood circulation. Because of this, GBMs can be promising in the development of drug carriers [23,24]. The unique properties of GBMs, such as high biocompatibility, electrical conductivity, mechanical properties, elasticity, antibacterial properties, and potential surface modification, can be applied for nerve tissue engineering (NTE) and regeneration [25,26,27,28]. GBMs enhance interactions between neurons and support neural tissue regeneration by acting as a bridge between regenerating neurones and retaining electrical conduction between distal and proximal neurones [29,30].

The application of stem cells is a critical parameter to be considered in NTE [2]. Previous studies have demonstrated that induced pluripotent stem cells (iPSCs), neural crest stem cells (NCSCs), and mesenchymal stem cells (MSCs) promote nerve regeneration by differentiating into Schwann-like cells and secreting neurotrophic factors [1,31]. For example, despite low differentiation of adipose-derived stem cells (ADSCs) in clinical studies, the conversion of neuron-like cells reached about 90% in the GO mat (glass coated with GO) [32]. GBMs are still new in NTE, and there is a need to explore their full potency in this field. This paper discusses the different aspects and characteristics of GBMs, reviewing the last six years of literature in the application of GBMs in NTE in its various forms and structures.
Figure 1Introduction of nervous system. (Illustrated by authors and utilising a real image of damaged myelin [33]).
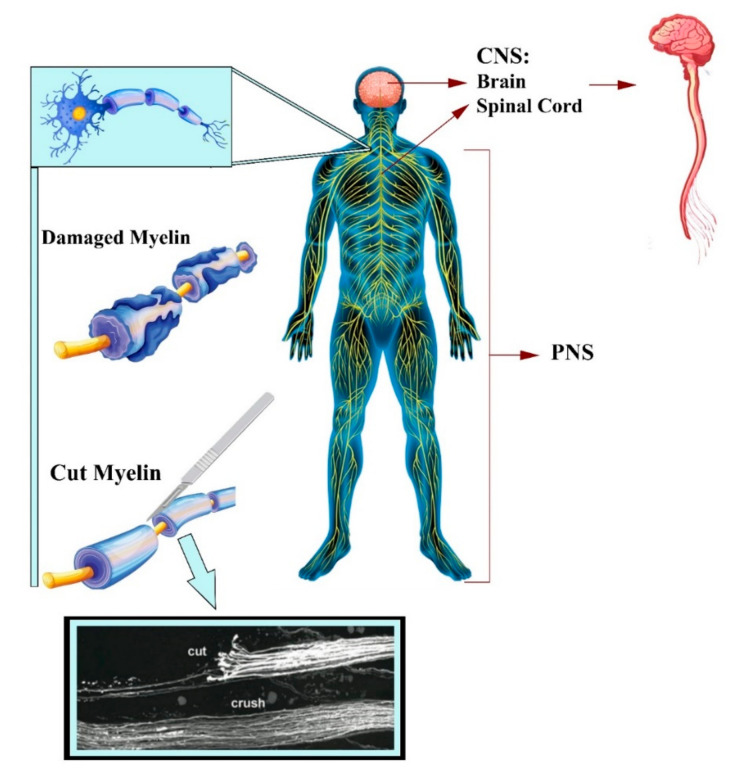

Figure 2Due to the damage, the axons and myelin are fragmented at the injury site. Dedifferentiation and proliferation of mature myelinating Schwann cells occur. Then, after dedifferentiation, myelin and axonal debris are removed by Schwann cells or by recruiting circulating macrophages and producing neurotrophic factors that support axon regeneration. Schwann cells downregulate myelin-associated genes, which are vital for myelinations such as Krox20/Egr-235, and re-express molecules correlated with immature states such as the p75 neurotrophin receptor (p75NTR) and the neural cell adhesion molecule (NCAM). Reused with permission [10].
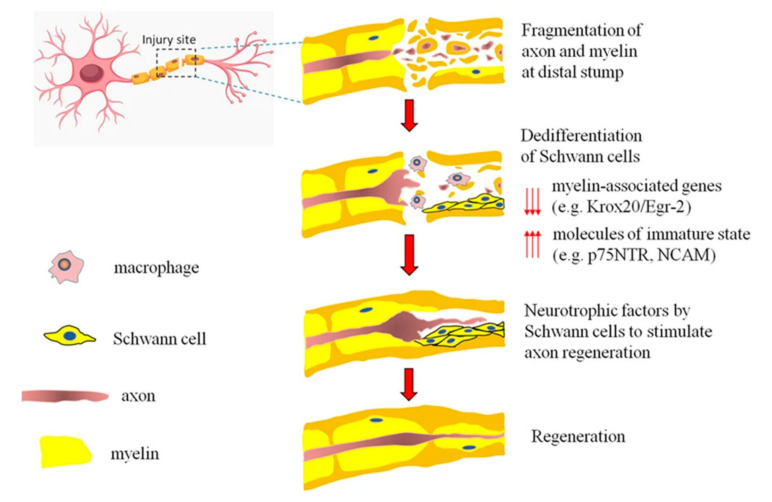

Figure 3Schematic of graphene (G), graphene oxide (GO), reduced graphene oxide (rGO) and functionalized graphene (FGO) materials in nerve tissue engineering, both in vitro and in vivo, in various forms of scaffolds, such as film, electrospun mat, foam or sponge, hydrogel, 3D print, and conduit.
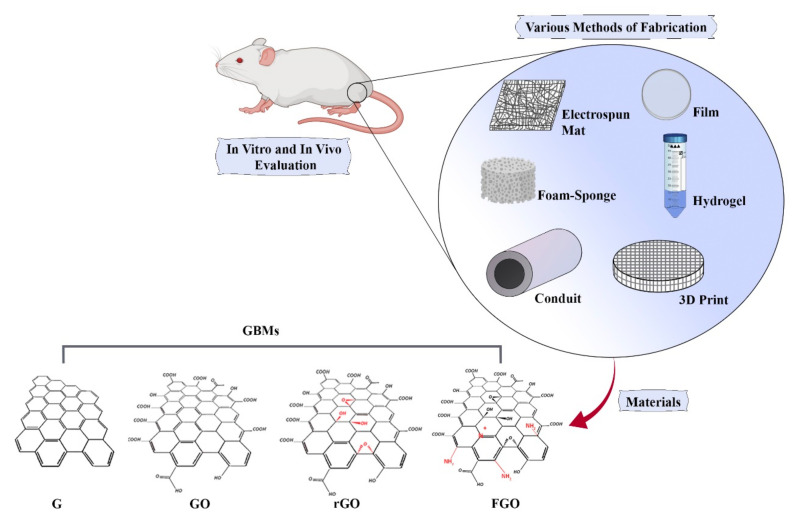



## 2. Tissue-Engineered Scaffolds with GBM and Neural Tissues

There are several scaffolds that can be applied to nerve repair and regeneration of the CNS and PNS. At present, there is more evidence to support the treatment of PNI secondary to the PNS high regenerative capacity after damage, in contrast to the CNS [34]. For NTE to progress, further research is required investigating the optimum parameters to develop tissue-engineered scaffolds that substitute the grafting procedure. The main challenges to improve neural regeneration involve bypassing the harsh microenvironment (e.g., inflammatory, and inhibitory) created following nerve injury. Following trauma or neurological disease, the myelinated fiber tracts are damaged gradually and the blood cells cross the broken BBB, invading the medullar tissues that lead to inflammatory responses. The excitatory neurotransmitters accumulation and inflammatory cytokines cause formation of an inhibitory extracellular matrix (ECM) [35,36]. Chemical signals, such as bacterial endotoxins and cytokines, direct macrophage inflammatory (M1) and M2 phenotypes [37]. M1 promotes cell recruitment and proliferation, while M2 encourages differentiation. The polarization phenotype, switching from M1 to M2, is influential in promoting functional tissue regeneration [38]. Therefore, TE scaffolds can be applied to stimulate endogenous nerve migration and control inflammatory cell infiltration.

GBMs have recently gained a lot of attention as promising candidates for the treatment of nerve injury and nervous tissue regeneration. Beneficial properties of GBMs include supplying electrical conductivity, which enhances mechanical strength and cellular behavior. The GBMs incorporation has been shown to improve the mechanical properties of bio-composites; higher concentrations of GBMs result in more suitable mechanical properties, which are adjustable by concentration for the target tissue [39,40,41]. In addition, the hydrophobicity of NTE structures enhances with increasing G amounts, which affects nerve cell attachment, proliferation, and differentiation. Further research should focus on ideal concentrations of G to achieve optimal results [15,31,42].

This paper focuses on the effects of GBMs and their potential for NTE. G-based scaffolds are categorized based on their structure: electrospun fiber, film (membrane), hydrogel, foam, 3D printed or bioprinting, and conduit (Figure 3), which are listed in Table 1 and Table 2. There are two concepts for GBMs in developing TE scaffolds promoting nerve regeneration: (1) evaluating the effect of GBMs concentration on characteristic outcome (prominently with electrospun fiber s) and the (2) addition of a certain amount of GBMs to optimize the scaffold by enhancing the mechanical strength, cellular behavior, and electroconductivity (commonly nerve guidance conduit and hydrogel).

### 2.1. Electrospun Fibers for Manufacturing 3D Scaffold

Electrospinning is a well-known fabrication method for biomedical application [43], and electrospun nanofibers are suitable candidates for NTE due to their 3D structure, flexibility, and similarity to neurites [24]. However, there are some debatable parameters in fabricating the best electrospun mat and membrane for nerve regeneration, which will be discussed in this section. Adjusting the concentrations of G added to the GBM-containing nanofibers allows for appropriate modification of its properties. Thus, achieving the optimum amount of G concentration is crucial. One study investigated the effect of rGo concentration (0.05 to 0.2 mg/mL) coated on polycaprolactone (PCL) fibers. The topography results (Figure 4A) revealed that as concentrations increased, rGO aggregated gradually on the surface of the PCL fibers, and they were no longer smooth surfaces. This study revealed the concentration of less than 0.1 mg/mL of rGO resulted in nil detection of conductivity due to insufficient coverage of rGO on the PCL electrospun mat. Thus, the 0.1 mg/mL rGO was selected as the optimum concentration to be coated on the PCL nanofiber [44]. Another study investigated the effect of rGO concentration (1% to 10%), which was incorporated in the silk fibroin electrospun mat. The rGO concentration did not influence the fiber diameter, overall porosity, and water absorption, and, in contrast, the surface roughness increased directly based on rGO concentration (Figure 4B). It is important to note that the presence of a hydrophilic polymer or selecting a polymer with a large absorption capacity (e.g., silk fibroin) leads to an increase in cell attachment [45]. Thus, as an alternative to conventional nanofibers, the wet-electrospun microribbons can be more effective in cell attachment due to the higher aspect ratio and canal shape morphology. The 1% wt G concentration resulted in slightly larger grooves with an average size of 1 μm. In addition, the presence of G alongside poly(lactic-co-glycolic acid) (PLGA) can provide oxygen functional groups and improved hydrophilicity. The complementary analyses revealed that incorporation of 1% wt G increased both conductivity (from 0.15 ± 0.01 μs/m to 0.42 ± 0.03 s/m) and elastic modulus (from 5.04 ± 0.5 to 17.1 ± 1.2 MPa) in comparison to the neat PLGA fiber [46]. Although, the increment of G concentration up to 20% wt exhibited no apparent cytotoxic effect, the cellular analysis revealed that Schwann cells were grown higher in 10% G concentration rather than 15% and 20%. In addition, such a high concentration in the electrospinning method might block the needle, and liquid droplets occur [47].

GO or rGO can be coated on particles such as Au NPs to increase surface functionalization and decrease the adverse effects of residual chemicals related to NPs synthesis. The size of NPs typically selected is below 100 nm to increase interaction with cells, even at the fundamental molecular level [48]. The concentration of 0.005% of rGo-Au NPs leads to a one-fold increase in neurite growth. The electrical conductivity of rGo-Au NPs increased from 157.5 μs/cm (Au-NPs) to 384 μs/cm, which referred to the removal of excess oxygen ions and made electron transportation ultrafast in rGO with an inherent electron mobility limit [48]. In another study, the rGO-Au NPs were incorporated into the polyhydroxyl alkanoate (PHA) fibers, and the study endorsed the SCs proliferation as well as migration under applying 100 mv/cm electrical stimulation (ES) [49].

The majority of studies on NTE have used ES to promote axonal regeneration in both the preclinical and clinical stages [50]. By applying ES, the cell membrane is subjected to a degree of depolarization, and significant depolarization can induce cell migration, proliferation, and axonal growth by modulating the distributions of ion channels (Figure 5) [6,45,51]. It is evident that increasing G concentration increases the scaffold’s conductivity. Additionally, increasing the voltage up to 100 mv/cm results in more axons of PC12 cells sprout [7]. Wang et al. demonstrated that the presence of 100 mv/cm causes not only perfect in vitro analyzes (Schwann cell migration, proliferation, myelin gene expression, neurotrophin secretion, and induced PC12 cell differentiation) but also promotes peripheral nerve repair in vivo at a similar level to the gold standard autograft [49]. Although most studies applied an intensity of 100 mv/cm [7,49,52], less intensity also leads to acceptable results. For instance, applying ES with an intensity of 50 mv/cm for one h/day through sodium dodecyl benzenesulfonate (DBS)/GO/polypyrrole-poly-l-lactic acid (PLLA) nanofiber can significantly promote neurite elongation and alignment [53]. Another study represents that significant intensity of ES may damage the neural cells and tissue, thus fabricating electrical responsive scaffolds that can deliver electrical signals to neuron cells and modulate cellular behavior for developing functional connection is highly recommended [32,52,54]. Applying a nominal ES intensity of 10 mv (1 h/day) alongside a superior conductive scaffold (4% G/thermoplastic polyurethane) with conductivity of 33.45 ± 0.78 S/m would be an ideal candidate for guiding neural cell growth [32]. However, applying negligible intensity of ES demonstrated a positive effect on cell proliferation and differentiation; some researchers believe in fabricating self-electroactive nanofibrous scaffolds without requiring ES [45,55].

A number of studies have demonstrated that aligned nano and micro-electrospun fibers can facilitate cell spreading, avoiding aggregation, and lead to improved cell migration and proliferation [44,56,57]. Furthermore, the anisotropic characteristic of the aligned fibrous scaffold can significantly increase the mechanical properties along with cell-scaffold integration, compared with random fibrous orientation [58]. Studies revealed that aligned electrospun fibers can conduct neonatal nerve capillary growth as well as fiber orientation to support functional nerve regeneration [46]. This environment is an ideal candidate for peripheral nerve regeneration due to its morphological resemblance to axons [44]. Various studies have demonstrated the importance of the similarity of nanofibers orientation to the ECM [7,27,45,47,48,59,60,61,62,63]. A minority of researchers neglect this parameter referring to orientation, which is possibly secondary to a lack of equipment for the electrospinning of aligned nanofibers. Cell type is a significant parameter for electrospun mats and is often controversial with selection. In addition to the mentioned stem cells in the introduction section, in some nanofibrous mats research, cellular analyzes were repeated with different cell lines to validate their results. The most frequently used cells are PC12 and Schwann cells [7,49,52,61]. Subsequently, their results can prove more reliable for future or even clinical studies.

Like the electrospun fibers or mats, membranes (films) can be helpful in particular NTE applications. The membranes are usually produced by the solvent casting method, and the G, GO, or rGO concentration selected is typically between 0.1% and 1.5% wt to achieve the best results in electrical conductivity, membrane flux, mechanical characteristics, and cellular behavior [64,65,66]. Another study investigated the long-term biological characteristics of graphene-based nanoscaffolds (GBN) in the PNS. To this aim, the GBN was manufactured using a layer-by-layer casting technique. The results indicated that a low concentration of GBN (4% G in the PCL scaffold) might be biocompatible since it exerts no appreciable toxicity in sensitive tissues such as liver, kidney, lung, heart, and spleen in the long-term repair (18 months) of peripheral nerves in vivo. The manufactured scaffold had biologically regenerative effects on myelination, axonal outgrowth, and locomotor function recovery [67].

### 2.2. Hydrogels

Hydrogels are 3D networks containing crosslinked hydrophilic polymer chains that can absorb a large amount of water; it is this characteristic that allows them to mimic the natural ECM [68]. Hydrogels are remarkably promising for application of the sustained delivery of biomolecules and can be directly implanted defect site, as well as for TE applications [12]. Hydrogels are the most promising choice for the delivery of genes or cytokines for nerve proliferation and differentiation. The addition of GBMs to hydrogels has gained recent attention due to its desired resultant properties: (1) higher stability and controlled delivery of molecules, GO in particular; (2) overcoming the notable weak point of hydrogels, and low mechanical properties, by acting as a capable reinforcement; and (3) creating conductive hydrogels that are highly useful for mimicking the natural environment in the nervous tissues or organs [69,70,71,72,73].

There are different aspects to investigate the effects of GBMs on hydrogels for NTE, such as the effect of concentration, pore size, morphology, chemistry (related to the functional groups), mechanical strength (elastic modulus in particular), and electrical conductivity [69,70,71,72,73]. GBMs are a candidate as a carrier for gene delivery and cell transfection, which can be better controlled and sustained within hydrogels [58,63]. In a study focusing on promoting BMSCs’ recruitment and stimulating sensory nerve regeneration, the stromal cell-derived factor-1α (SDF-1α), a member of the chemokine family of pro-inflammatory mediators, and pDNAs were used. To this aim, 25 kDa polyethylenimine (PEI) was conjugated to nanoscale GO sheets to deliver pDNAs encoding bFGF (GO-PEI-bFGF) and crosslinked by matrix metalloproteinase (MMP)-2. The crosslinking of GO-PEI sheets keeps the DNA inside to prevent degradation and induce the responsive BMSCs transfection. This gene delivery system was encapsulated by a GO-based hydrogel that contains SDF-1α. SDF-1α can recruit distant endogenous BMSCs, and MMP-2 activated the hydrolysis of crosslinked GO-PEI and then started the BMSCs transfection towards the neural-like cells (Figure 6A). Using GO resulted in several improvements, such as stable biomolecule delivery and release in a controlled and sustained way. Although there are still some concerns regarding GO biocompatibility, this study confirmed the dose-dependent cytotoxicity of nano (significantly >10 μg/mL) and microscale (even noticeable at 1 μg/mL) GO. Conjugating the cationic PEI to the GO surface improved GO biocompatibility remarkably (GO-PEI 1:10 was the optimum) [69]. Gene delivery with the help of GBMs promises hope by maximizing therapeutic efficacy, the substantial factor to consider for delivery efficiency with gene transfection, and according to the reports; this type of engineered delivery system can introduce treatments for CNS diseases [69,74].

To evaluate the impact of GBMs on the electrical conductivity of hydrogels, GBMs can be applied solely, with a combination of other conductive biomaterials or even with the addition of electric charges, to result in an appropriate range of conductivity. Introducing electric charges is a strategic method to stimulate the proliferation and differentiation of nerve cells [72,75,76]. This is because excitable cells rely significantly on electrical conductivity, and ion accumulation and flow, to coordinate cellular functions and signal transduction [75,77]. In order to form a positively charged hydrogel, oligo(poly(ethylene glycol) fumarate) (OPF) was crosslinked with [2-(methacryloyloxy)ethyl]trimethylammonium chloride (MTAC). The results have shown enhanced neural cell adhesion, proliferation, and differentiation. Furthermore, from the dissociated embryonic chick dorsal root ganglion (DRG) explant, it can be said that OPF hydrogels resulted in a combination of neurons, neuronal support cells, and Schwann cells [75]. A study depicted that the introduction of positive surface charges (OPF and OPF-MTAC hydrogels) to conductive carbon components (GO and carbon nanotube, CNT) can remarkably stimulate nerve cell responses [72].

Using anti-inflammatory drug is an alternative approach to enable nerve regeneration that can be combined with loading GBMs. A hydrogel was engineered using GO sheets with four-armed polyethylene glycol and functionalized with diacerein as an M2 drug (4arm-PEG-diacerein or PD). The designed hydrogel was injectable and self-recovery due to the strong physical interactions between GO and diacerein. The range of conductivity of the PD/GO hydrogel was consistent with suitable conductivity for NTE materials [78,79]. It has been reported that a suitable range of electric conductivities is 1–10 S m^−1^, which results in neuron growth, longer neurite length, faster neurite growth rate, and better axon remyelination [78,79]. Further, PD/GO hydrogel provided an anti-inflammatory microenvironment; synapses appeared around the cells. Therefore, this hydrogel can promote neuronal network formation at the cellular level and inhibit subsequent hyperactivation of astrocytes caused by reactive oxygen species (ROS). In addition, PD/GO hydrogels remarkably reduced the lesion area and the site of inflammation compared to the others (Figure 6B) [70]. Other approaches similar to hydrogel, such as aerogel and foam, are also considered common methods in fabricating high porous materials for biomedical applications. For instance, a G foam/hydrogel scaffold has been utilized for peripheral nerve regeneration. The in vitro results indicated that ADSCs can regulate Nrf2/HO-1, NF-κB, and PI3K/AKT/mTOR signaling pathways, showing multiple functions in reducing oxidative stress and inflammation and regulating cell metabolism, growth, survival, proliferation, angiogenesis, differentiation of Schwann cell, and myelin formation. Furthermore, the in vivo results demonstrated ADSCs-loaded composite scaffold significantly promoted nerve recovery and inhibited the atrophy of targeted muscles [80]. Likewise, some recent cases with GBMs and these techniques (aerogel and foam) for NTE have been listed in Table 1, although they are limited at present when compared with other techniques.

### 2.3. 3D Printing and Bioprinting

Both three-dimensional (3D) printing and bioprinting are counted as the additive manufacturing process that can print precise complex design to repair damaged tissues or organs; however, the main difference between these two methods is that 3D bioprinting builds customized structures from cells and supporting biomaterials (bioink), while 3D printing solely uses biomaterials for printing [12]. Although there is a high volume of research on 3D printing and bioprinting, the literature on their applications in NTE is still relatively limited. The addition of GBMs can lead to a higher viscosity (due to the shear thinning properties), hence improving printability. In addition, water-dispersible GBMs can also be used as a component of bioink; however, the concentration plays a significant role and therefore needs careful evaluation [15,81,82,83]. A study was performed using waterborne biodegradable polyurethane with soft segments that mainly included poly(ε-caprolactone) and 20 mol% of shorter poly(D,L-lactide) chains. The polyurethane dispersed (25 ppm) in a cell culture medium then underwent a sol-gel transition near 37 °C with a proper gel modulus. Afterwards, G or GO was mixed with polyurethane to prepare a G-based bioink for neural stem cell printing, which resulted in a suitable bioink for printing and cell survival (Figure 7). Interestingly, the addition of G or GO at a very low content (25 ppm) not only resulted in a promising bioink but also significantly improved oxygen metabolism (2- to 4-fold increment) and neural differentiation. On the basis of these results, it can be concluded that the optimum sample was PU/G since it presented better efficacy, especially for cell proliferation and differentiation and oxygen metabolism [69]. It is worth mentioning that in order to achieve these precise conduits, 3D printing can be applied alongside the other techniques, and the related studies are given in the conduit section [25,28,83]. Table 1, highlight the recent studies in application of graphene based materials for nerve regeneration. 

**Table 1 biomedicines-10-00073-t001:** Tissue-engineered 3D scaffolds containing graphene-based materials with for nerve regeneration. **Keywords:** ApF, A.pernyi silk; BM-MSCs, bone marrow-derived mesenchymal stem cells; CNS, central nervous system; CNTpega, Carbon nanotube poly-(ethylene glycol) acrylate; CPM, cell proliferation and migration; MP, mechanical properties; MTAC, 2-(methacryloyloxy)ethyltrimethylammonium chloride; NGF, nerve growth factor; NR, nerve regeneration; NSCs, neural stem cells; OPF, oligo(poly(ethylene glycol) fumarate; PCL, polycaprolactone; PEG, polyethylene glycol; PEI, polyethylenimine; PLGA, poly(lactic-co-glycolic acid); PLLA, polypyrrole-poly-l-lactic acid; PNS, peripheral nerve system; PVA, polyvinyl alchol; RGCs, retinal ganglion cells; SCI, spinal cord injury; =>, result in; +, by addition; ↑, higher or increase; ↓↓ minimize; * tested in preclinical rat model.

	Biomaterial(s)	GBMs Concentration	Target and Cell Type	Outcomes	Year
**Electrospun fiber**	Silk/rGO and SF/rGO (Post reduction of Silk/GO)	5% and 10%	PNS Neuronoma NG108-15	Conductivity: 4 × 10^−5^ S/cm (dry), 3 × 10^−4^ S/cm (hydrated) ↑ metabolic activity and CPM in SF/rGO Neurite extensions up to 100 μm	2021 [45]
	Polydopamine/carboxylic GO/PLLA (PDA/CGO/PPy-PLLA)	0.03% wt	PNS Schwann cells	Surface conductivity: 17.35 S/m Elastic modulus: 260 MPa, ↑ CPM ↑ neural proteins expression 50 mV/cm => ↑∼31% of Schwann cells to align along the direction on mat	2020 [63]
	PCL/G	1% and 2%	NTE Mouse E12	↑ concentration => ↑ fiber diameter, elastic modulus, max stress, and differentiation	2019 [84]
	* GO/ApF/PLCL	1–2 mg/mL	Sciatic nerve repair, Schwann cells and PC12	Optimum (mg/mL): 2 ↑ CPM, ↑ PC12 differentiation and FAK expression, ↑ myelination, and repaired 10 mm sciatic nerve defect	2019 [61]
	G/PVA	1%	PNS PC12	Orientation Index: aligned scaffold: 28.7° and native nerve: 26.8°, hydrophile, strength: aligned: 29.6 ± 6.7 MPa, + ES => ↑ CPM (aligned > random)	2018 [58]
	G/Sodium alginate/PVA (G/AP)	(0.5–5)% wt	NTE PC12	Optimum: 1%, ↑ concentration => ↑contact angle, degradation, conductivity (1%): 800 μs, ↑ CPM (1.4 times in 1%)	2017 [62]
	Polypyrrole/G/PLGA (PPy-G/PLGA)	1 and 6 % PPy-G	Optical NR (Glaucoma) RGCs	Well aligned, +ES => ↑ cell length ↑ cell viability and neurite outgrowth Anti-aging ability of RGCs	2016 [57]
**Hydrogel**	* Grafted GO/PEI Core–shell microfiber arrayed hydrogel: chemokine (SDF-1α) and GO-PEI/pDNAs-bFGF microparticles	Mass ratio (GO-PEI): 1:10, 1:40, and 1:70	Recruit and stimulate the neural-like differentiation BM-MSCs	Optimum ratio: GO-PEI 1:10, ↑ neuronal differentiation, controlled delivering the CXCL12 and GO-PEI/pDNAs-bFGF => endogenous stem cell therapy	2021 [69]
	* GO/diacerein-terminated 4-armed polyethylene glycol	2.5, 5.0, and 7.5 mg/mL	SCI	Optimum concen. (mg/mL): 5.0, Diacerein => ↓↓ inflammatory response and ↓↓, inhibitory microenvironment, conductivity (7.4 S/m) => ↑ neuron growth and axon remyelination	2020 [70]
	Polyacrylamide/GO/gelatin/sodium alginate (PAM/GO/Gel/SA)	0.5% and 1%	PNS Schwann cells	Optimum %: 0.5, ↑ protein adsorption ↑ NGF, ↑ cytoskeleton related genes expression	2018 [71]
	GO acrylate (GOa) and CNTpega embedded in OPF hydrogel MTAC => rGOaCNTpega-OPF-MTAC	0.1% *w*/*v*	Neuronal proliferation and differentiation PC12 cells	Conductivity: 5.75 × 10^−3^ S/m ↑ differentiation (+NGF) Robust neurite development Conductive nerve conduits with surficial positive charges	2017 [72]
	GO/polyacrylamide	(0.5–3)% *w*/*v*	PNS Schwann cells	Optimum %: 0.4% GO, ↑ biofactors release and larger matrix adsorption	2016 [73]
**Aerogel**	Hollow GO/gelatin	5 mg/mL	NR and CNS P19 mouse cells	Prevent the fibroglial tissue formation Effective differentiation into neural cells	2020 [85]
	GO/SA and rGO/SA	0.5, 1, 3, and 5 mg/mL	CNS NI	↑ porous and electroconductive Similar MP to CNS (↑ rGO)	2019 [86]
**Foam**	G foam/PCL mesoporous coating	1–7.3 wt%	NTE NI	Conductivity and MP (Young’s modulus): 3.2–108.7 S/m and 0.62–4.50 MPa	2021 [87]
**Bioprinting**	PU/G or PU/GO (G and GO coated by Pluronic P123)	10, 25, and 50 ppm	CNS NSCs	Optimum ppm: 25 PU/G > PU/GO: Suitable cell survival rate ↑ CPM ↑ differentiation ↑ oxygen metabolism and ATP production	2017 [81]
	gelatin methacrylamide (GelMA)/G	1 mg/mL	NR PC12 cells (biocompatibility) and NSCs (cell encapsulation)	↑ CPM ↑ neuron differentiation and neurites elongation	2016 [82]

### 2.4. Conduits for Nerve Guide

When the direct end-to-end tensionless repair for the injured nerve is not possible (>30 mm gap), the first choice for management would be to perform autologous nerve grafting [88]. Autologous nerve grafting is considered to be the gold standard treatment for PNI; however, the procedure has multiple complications such as a shortage of donors, donor-site morbidity (e.g., sensory loss and neuroma formation), and disease transmission [89]. The development of a polymer-based nerve guidance conduit (NGC) has made considerable progress to encourage nerve repair and regeneration in a targeted manner with their physical and chemical features; however, there is still a gap to achieve structural and functional restoration that mimics the natural properties. Current research supports the use of NGCs to repair large nerve defects, particularly in the PNS [90].

NGCs are used to bridge the nerve stumps and eliminate scar formation, help the axon stretch, and form a suitable microenvironment to regenerate the injured nerve [91]. NGCs require great mechanical strength, as well as adequate elasticity to allow regular muscle motions around the conduit without tube collapsing [72]. The recovery is extremely effective as nerve extension is hampered through the generation of scar tissue during the repair process [92,93]. Injuries related to the PNS are common (13−23 out of 100,000 individuals are effect annually in developed countries [94]); so, research has been focused on developing appropriate NGCs through several techniques to improve treatment. To meet the various needs of a suitable NGC for nerve regeneration, “multifunctional NGCs” are needed; hence, multiple biomaterials, biomolecules, and fabrication techniques have been used in most case studies [25,59,93,95,96].

The properties of GBMs mean that they are promising biomaterials for the successful fabrication of NGC. One successful example includes an NGC synthesized by a 3D graphene mesh tube (GMT) and subsequently filled with alginate-gelatinmethacryloyl (GelMA) hydrogel, which was also evaluated by animal studies (Sprague–Dawley rats). The addition of alginate enhances the mechanical properties through a double network structure and supports tube formation. To enable axonal guidance and neurons migration, Netrin-1 (100 ng/mL) has been loaded (Figure 8A) [95]. The presence of G not only resulted in the enhanced proliferation of Schwann cells and guided their alignments but also led to a satisfactory Young’s modulus (725.8 ± 46.52 kPa) and electrical conductivity (6.8 ± 0.85 S/m). The release of netrin-1 was significant in directing axon pathfinding and neuronal migration, which optimized tube formation ability at 100 ng/mL. In vivo studies showed that the NGC successfully supported peripheral nerve regeneration, restored denervated muscle, and was superior to the positive control (autologous graft). Additionally, the revascularization of denervated muscle was achieved, which is a crucial factor for regeneration and recovery after PNI. Another achievement of this study was enhancing the survival and function of Schwann cells both in vitro and in vivo. The extraction and cultivation of Schwann cells is complex and inconvenient, so many studies concentrate on stem cells such as adipose-derived stem cells (ADSCs) that can differentiate into Schwann-like, neuron-like, and endothelial cells [95].

A further study utilized conductive hydrogel-based NGCs, made of GelMA/GO and followed by chemical reduction, to improve the electrical properties of the hydrogel, r(GO/GelMA) [97]. This is because restoring sp^2^-carbon bonds while minimizing rGO aggregation can enhance the electrical characteristics of GO-loaded hydrogels [15]. The fabricated NGCs had adequate electrical conductivity (r(GO/GelMA)>GO/GelMA), flexibility, mechanical strength, and permeability, and r(GO/GelMA) showed higher neuritogenesis enhancement of PC12 neuronal cells compared to previous studies. Muscle regeneration after NGC implantation was investigated by Sprague–Dawley rats by calculating the weight of the gastrocnemius muscles (both left and right). The left gastrocnemius muscle showed serious atrophy at 4 weeks, and the GelMA group underwent considerable shrinkage by 8 weeks. However, the other groups did not display atrophy and shrinkage; instead, they exhibited muscle recovery, particularly the r(GO/GelMA) and autograft (Figure 8B) [97].

In order to optimize and evaluate the G concentration, a collagen-based NGC based was prepared using the cryogel technique to regenerate the spinal cord. The cryogels were stabilized by using amine-functionalized graphene as a nano-crosslinker and resulted in super-macroporous cryogel (Figure 9A). With the addition of 0.1%, 0.5%, and 1% *w*/*v* of G, an upward trend in electrical conductivity was recorded; however, there was no further increment despite enhancing the concentration above 1% *w*/*v*, so this was found to be the plateau for the greatest amount of conductivity. The differentiation of rat bone marrow-derived mesenchymal stem cells (BM-MSCs) into neuronal-like cells has been demonstrated in the presence of graphene and ES. Based on the cell studies and organotypic spinal explants on samples, optimum neuronal differentiation was seen with 0.5% *w*/*v* G/collagen cryogels. Research has shown that BM-MSCs-seeded cryogels were able to secrete ATP energy upon ES. 0.5% *w*/*v* crosslinked collagen cryogels supplied adequate mechanical and electrical cues that encouraged the significant extracellular secretion of ATP. The cryogels have also shown that having a mixed population of M1 and M2 macrophages is necessary for nerve tissue repair [98].

The introduction of micropatterns and bioactive substances into the inner wall of NGCs play a significant role in regulating Schwann cells behavior, axons elongation, and macrophages phenotype. Linear micropatterns with various ridges and grooves (3/3, 5/5, 10/10 and 30/30 μm) were made on poly(D,L-lactide-co-caprolactone) (PLCL) films; following this, surface aminolysis and GO electrostatic adsorption were conducted (Figure 9B). GO has been used to provide enhanced cell attachment and proliferation, as well as electrical conductivity, in turn improving guiding. The GO-modified micropatterns accelerated the collective migration of Schwann cells and directed cells along with stripes by the fastest rate on the 3/3-GO film that contains the largest force of cell adhesion. It also resulted in tending macrophages to differentiate into the M2 type on the 3/3-GO film (optimum NGC). The NGCs were implanted to bridge the 10 mm rat sciatic nerve defects for 4 and 8 weeks. To investigate function recovery of regenerated sciatic nerves, at 4- and 8-weeks post-operation, nerve conduction velocity and compound motor action potential were measured. The 3/3-GO group represented higher values than other groups. Additionally, in the 3/3-GO group, myelinated nerve fibers and blood vessels were generated more significantly than others after 8 weeks [99].

**Table 2 biomedicines-10-00073-t002:** Studies related to the nerve guidance conduit development based on graphene-based materials. Keywords: BM-MSCs, bone marrow-derived mesenchymal stem cells; SC, stem cell; NR, nerve regeneration; CNS, central nervous system; PNS, peripheral nerve system; SCI, spinal cord injury; SDR, Sprague–Dawley rat; PNR, peripheral nerve regeneration; CPM, cell proliferation and migration; RGD, arginylglycylaspartic acid; PCL, poly(ε-caprolactone), PVDF, polyvinylidene fluoride; PLA, polylactic acid; ApF, Antheraea pernyi silk fibroin; NI, not investigated; ↓, lower or decrease, ↑, higher or enhance; =>, result in; and +, by addition.

	Biomaterial(s)	GBMs Percentage	Target and Cell Type	Animal Model	Outcomes	Year, Ref
**Film** **(membrane)**	PLCL/GO	1 mg/mL	PNS Schwann cells	SDR	Elastic modulus: 125 MPa ↑ directional migration of single cells along the micropatterns. Macrophage differentiation into the M2.	2020 [99]
	PCL/carbon and G nanoparticles	0.5%	PNS NI	Lewis rats	↑ CPM ↑ Flexibility => ↑ stump positioning accuracy ↑ myelinated axons Muscle atrophy protection (12 weeks)	2017 [100]
**Electrospun fiber**	PCL/collagen/G	0.5%, 1%, 1.5%, and 2%	Sciatic nerve repair MSCs	SDR	Well aligned Optimum %: 1 ↑ concentration => ↑ conductivity; Conductivity (1%): 5.27 × 10^−6^ S/m ↑ concentration => ↓ tensile strength and elastic modulus	2020 [54]
	Dual-electrospun: PCL, gelatin, and polyaniline/G (PAG)	0–3% wt	PNS BM-MSCs	NI	Optimum: 2% PAG => ↑ CPM Conductivity: 10.8 × 10^−5^ S/cm	2020 [101]
**Foam**	G/PCL	2% wt	PNS PC12 cells	NI	Elastic modulus: 2.67 MPa Conductivity: 25 S/m ↑ porosity ↑ cell proliferation and extension No cytotoxicity	2021 [102]
	PVDF/GO	0.5%, 1%, 3%, and 5% wt	PNS PC12 cells	NI	↑ piezoelectricity and electrical conductivity High flexibility => easy and appropriate NGC formation ↑ CPM (particularly 0.5% and 1%)	2019 [103]
**Hydrogel**	GMT/ hydrogel with netrin-1	0.05%	PNS Schwann cells	SDR	Elastic modulus: 720 kPa Conductivity: 6.8 S/m GMT => orientation of PNR, O2 and nutrition transport. ↑ levels of S100 and Sox10 (↑PNR).	2021 [95]
	GO/GelMA then chemically reduced => r(GO/GelMA)	0.1%	PC12 cells	SDR	Conductivity: 4.4 × 10^−3^ S/m (GO/GelMA) and 8.7 × 10^−3^ S/m (rGO/GelMA) ↑ neuritogenesis: r(GO/GelMA) > GO/GelMA ↑ PNR ↑ regrowth with myelination	2020 [97]
	Chitosan/oxidized hydroxyethyl cellulose (CS/OHEC) hydrogel loaded with asiaticoside liposome and rGO	0%, 1%, 2%, 4%, 6%, 8%, and 10%	PNS PC12 cells	NI	Optimum: 8% Conductivity (5.27 × 10^−4^ S/cm) + ES => ↑ differentiation and ↑ CPM. Asiaticoside released => no growth and collagen secretion of fibroblasts => No scars for NR.	2020 [93]
**Cryogel**	Amino-functionalized G/collagen	0.1%, 0.5%, and 1% *w*/*v*	SCI BM-MSCs	Organotypic spinal explant culture (spinal cord from SDR)	Optimum %: 0.5% Conductivity: 3.8 × 10^−3^ S/cm and mechanical cues: 100–347 kPa Young Modulus => SC and NR ↑ ATP secretion ↑ MAP-2 kinase and β-tubulin III expression ↑ CD90 and CD73 gene expression	2021 [98]
**Multiple techniques**	Electrospinning, molding, and freeze drying: ApF/PLCL/GO	2%	PNS Schwann cells and PC12 cells (differentiation)	SDR	Effective guiding interface => ↑ CPM and ↑ myelination Tailored degradation and complete degradation at 12 weeks ↑ axonal regrowth and remyelination	2020 [96]
	Aligned electrospun and film: carboxylic GO-polypyrrole/poly-L-lactic acid (C-GO/PPy/PLLA)	0.05% *w*/*v*	PNSPC12 and L929 fibroblasts	SDR	↑ CPMConductivity: ~4.6 S/cm (after 4 weeks of immersion) + ES => re-innervated gastrocnemius muscle, nerve conduction, and neurite alignment (59%) at 12 weeks	2019 [104]
	Molding, phase separation (conduit), and 3D printing (circuit): Gelatin and G/PLA filament	Not reported	PNI MSCs	NI	+ ES => ↑ transdifferentiation into Schwann cell-like phenotypes ↑ CPM	2019 [83]
	3D printing-film: Polydopamine (PDA)/RGD and single-layered G (SG) or multi-layered G (MG)/PCL	1%	PNS Schwann cells	SDR	Conductivity: 8.92 × 10^−3^ S/cm (SG) and 6.37 × 10^−3^ S/cm (MG) Elastic modulus: 68.74 MPa (SG) and 58.63 MPa (MG) ↑ neural expression (SG>MG) ↑ axonal regrowth and remyelination	2018 [25]
	Molding/jet spraying/3D printing: GO/PCL	0.5%, 1%, 2%, and 4%	PNS Schwann cell	SDR	Optimum %: 1% Conductivity: 4.55 × 10^−5^ S/cm Elastic modulus: 48.32 MPa ↑ CPM Neural characteristics maintenance Angiogenic capability	2018 [28]

### 2.5. Associated Challenges of GBMs in Clinical Studies

The presence of GBMs caused conductive functionalities and high cell interactions. Among the various configurations of GBMs, NGCs have been shown to be the most promising candidate for the treatment of PNI since they can reduce axonal escape [105,106]. The most significant obstacle preventing further clinical studies with GBMs is the extensive safety assessment as GBMs are often identified as hazardous materials [107]. The major shortcoming identified in the ongoing studies arises from the utilization of simple cytotoxicity analyzes to evaluate the resultant scaffold’s toxicity, which leads to acceptable results but is insufficient for clinical use, so evaluation is highly recommended to validate results.

When developing a scaffold for NTE with GBMs incorporation, the concentration of incorporated GBMs is highly effective to the final properties. In some cases, the required concentrations of GBMs in TE are even greater than the loaded amounts of G in biosensors’ application. Therefore, the effects of GBMs concentration were thoroughly discussed and reviewed in this paper. For clinical implementation to take place, more research is required to study the long-term toxicity of GBMs, in which current literature remains limited [33]. In addition, to design an appropriate scaffold, which is promising for a clinical study, it is crucial to report on characterization analyzes such as physicochemical characterization, C/O ratio, surface area, cytotoxicity, genotoxicity, biodegradation, distribution, metabolism, and total accumulation in organs [15,33].

The development of GBMs for nerves is currently at preclinical animal model trials and may translate to the clinical setting within the next three to ten years. Our group in London is working on peripheral nerve regeneration using a novel biodegradable material (BioHastalex™), based on functionalized graphene oxide (FGO) covalently conjugated to the backbone of the chemical structure of polycaprolactone [108,109]. This work is currently at preclinical animal trials. The biocompatibility of BioHastalex™ and its unique biophysical and conductive properties have made it an attractive candidate as a material for nerve regeneration [110].

### 2.6. Conclusions

In conclusion, there is presently an unmet clinical need for the repair of transacted nerve injury with a gap >30 mm. The current commercially available decellularized or biomaterial-type nerve conduits fail to support neural regeneration in gaps >30 mm. Graphene and its derivatives are promising candidates in the treatment of nerve injury. Graphene is a single-layer atom, with superior mechanical and chemical properties, which include electrical and thermal conductivity, and strength (graphene is more than 100 times stronger than steel). There are several products currently under development at research centers as well as in industry for the development of nerve conduit from GBMs with biofunctionalization using stem cells and growth factors.

## Figures and Tables

**Figure 4 biomedicines-10-00073-f004:**
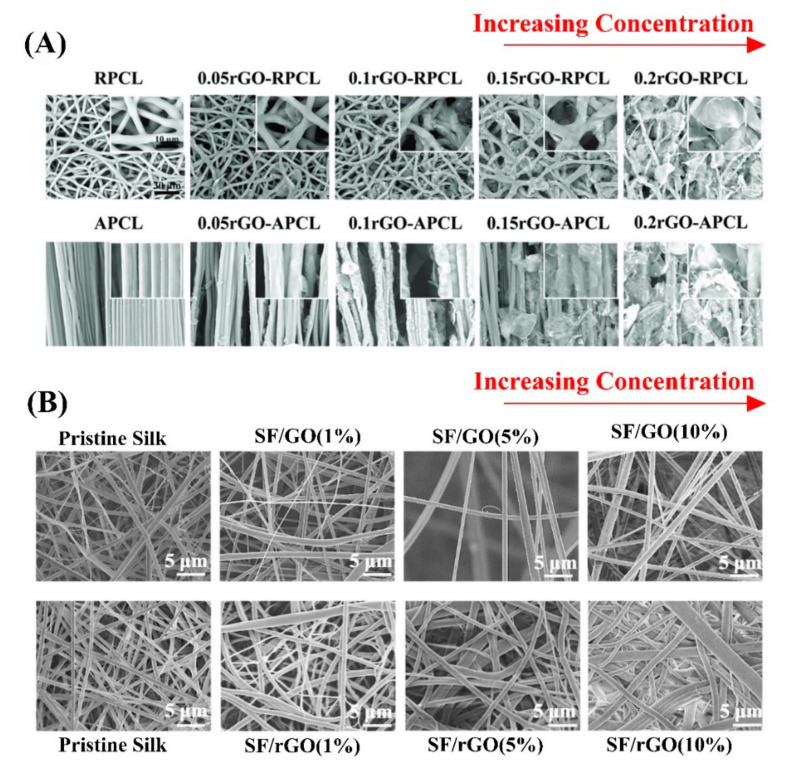
(**A**,**B**) exhibiting the morphology of electrospun mats by increasing the GBMs concentration. Reused with permission from [44,45].

**Figure 5 biomedicines-10-00073-f005:**
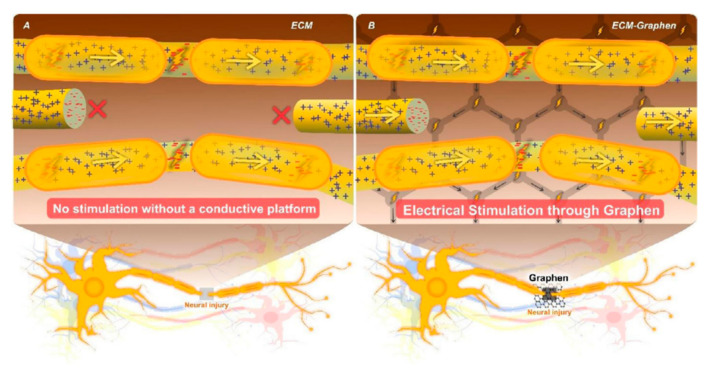
The schematic illustration of ES effect on neural injury regeneration (**A**) Injured neuron without the conductive platform and electrical stimulation, (**B**) Injured neuron exposed conductive platform and electrical stimulation. Reused with permission from [6].

**Figure 6 biomedicines-10-00073-f006:**
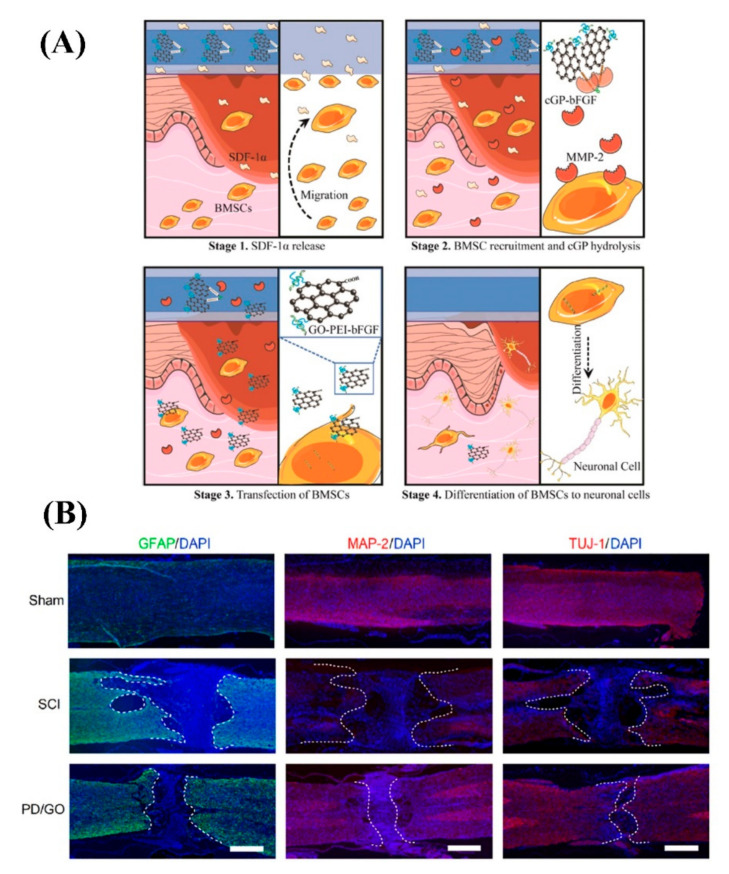
(**A**) Mechanism of the engineered system of GO-based gene delivery that induces differentiation of recruited BMSCs for cutaneous nerve regeneration. Reused with permission from [69]. (**B**) Immunofluorescence images indicate the structure of the injured spinal cords and the distribution of three important marker proteins: glial fibrillary acidic protein (GFAP), microtubule-associated protein 2 (MAP-2), and neuron-specific class III β-tubulin (TUJ-1). Scale bar: 500 μm. Reused with permission from [70].

**Figure 7 biomedicines-10-00073-f007:**
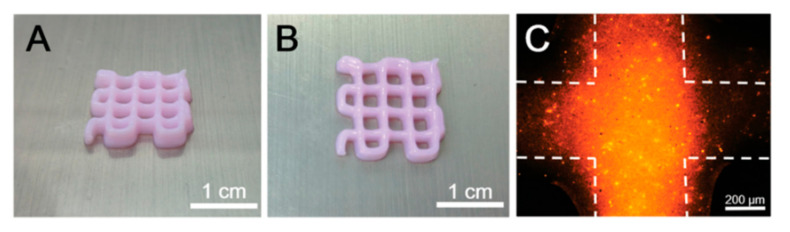
3D bioprinting structure made of PU/G. (**A**) Side view, (**B**) top view of the construct, and (**C**) image of neural stem cells encapsulated in the scaffold. Cells were labelled with PKH26 (red fluorescence). Reused with permission from [81].

**Figure 8 biomedicines-10-00073-f008:**
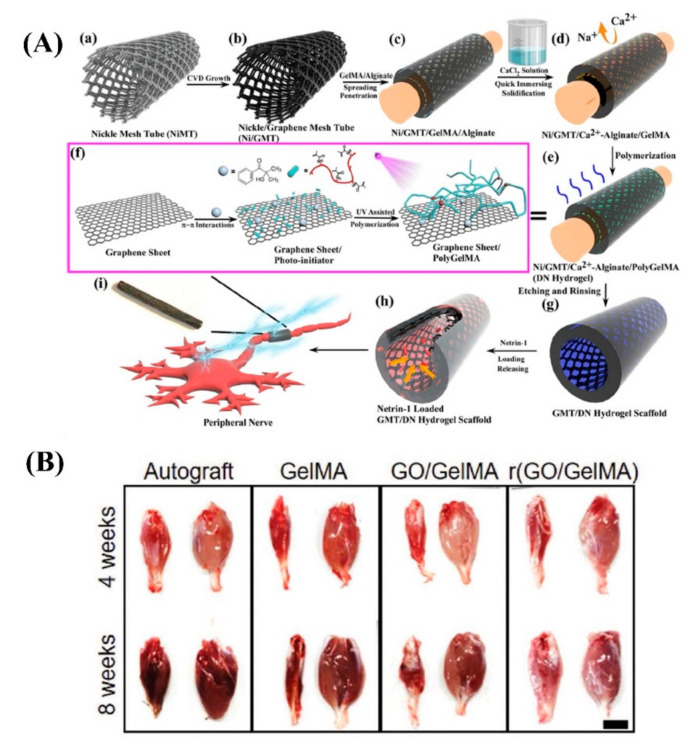
(**A**) Schematic of netrin-1-loaded GMT/hydrogel conduit preparation. (**a**,**b**) Growing G onto a nickel mesh (CVD method), (**c**) covering G/nickel mesh with a precursor solution, (**d**) formation of strong ionic bonds between alginate and Ca^2+^ ions due to immersing GMT into CaCl_2_ solution, (**e**,**f**) polymerization of GelMA under UV light, (**g**) etching nickel template, and (**h**) Immersing the conduit in high concentrated netrin-1 solution, (**i**) peripheral nerves regeneration. Reused with permission from [95]. (**B**) Images of regeneration of muscle after NGC implantation in different groups. Reused with permission from [97].

**Figure 9 biomedicines-10-00073-f009:**
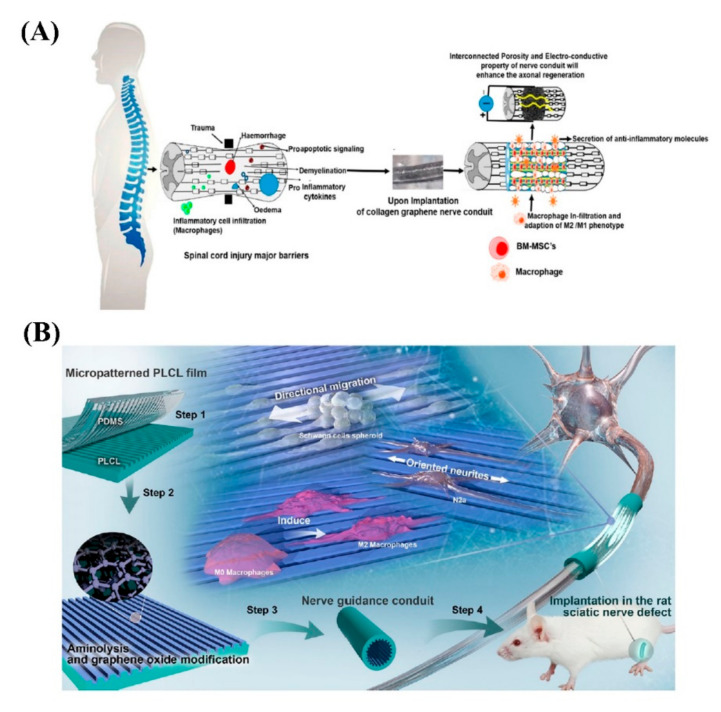
(**A**) Schematic of projected collagen graphene cryogel mechanism: due to spinal cord injury, inflammatory cytokines and infiltration of inflammatory cells have been produced. By implanting the cryogels, it will promote proliferation and stemness maintenance of BM-MSCs and secrete anti-inflammatory biomolecules. Further, the presence of cryogels and macrophage infiltration will stimulate high polarization of the M2/M1 phenotype. Reused with permission from [98]. (**B**) The illustration depicts the PLCL film fabrication with stripe micropatterns and GO nanosheets and its use in four steps: (1) creating micropatterns by thermal pressing of a polydimethylsiloxane template onto a PLCL film, (2) aminolyzing by 1,6-hexanediamine then GO adsorption electrostatically, (3) manufacturing micropatterned PLCL/GO conduit, and (4) implanting into a rat with sciatic nerve defects. The middle schematic shows that the micropatterned PLCL/GO film can improve the directional migration of Schwann cells from their cell spheroids, induce the macrophages differentiation into M2 type, and guide the neurites of N2a cells along with the patterns. Reused with permission from [99].

## Data Availability

Some data are available at www.NanoRegMed.com (accessed on 1 November 2021).

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
