# Peer review of "Graphene-Based Materials Prove to Be a Promising Candidate for Nerve Regeneration Following Peripheral Nerve Injury"

_biomedicines, 2021, doi:10.3390/biomedicines10010073_

Round 1

Reviewer 1 Report

This review summarize the state of the art of graphene-based biomaterials for PNS-nerve regeneration.  Some comments are below:

Line 36 (end)-40-page 2. This part is about CNS and drug carriers, it would be deleted because out of the focus of this review.

Typo line 14- page 3 Typo GBN

Line 6-9- page 4 can you add some refs here?

In general, the authors reports mainly in vitro tests for the Graphene-based different scaffolds. If this can be ok for fibers or hydrogel, maybe for the Nerve conduits they could try to report more studies of Graphene-enhanced conduits in vivo in animal preclinical models (if there are), or at least details more the in vivo results (also in a dedicated paragraph).

I understand from 2.5 that there are not clinical studies for graphene-enhanced scaffolds for PNS, is it correct?

Line15-20 page 16: this part should be in a par. About preclinical or clinical studies, not ion the conclusions. Maybe in the conclusions the authors should add their view of the next 3-5-10 years about graphene-based scaffold future.

Author Response

This review summaries the state of the art of graphene-based biomaterials for PNS-nerve regeneration. Some comments are below:

1- Line 36 (end)-40-page 2. This part is about CNS and drug carriers, it would be deleted because out of the focus of this review.

Response: Thank you for pointing out this. It has been revised. 

2- Typo line 14- page 3 Typo GBN

Response: Thanks, typo has been corrected.

3- Line 6-9- page 4 can you add some refs here?

Response: New references have been added.

4- In general, the authors reports mainly in vitro tests for the Graphene-based different scaffolds. If this can be ok for fibers or hydrogel, maybe for the Nerve conduits they could try to report more studies of Graphene-enhanced conduits in vivo in animal preclinical models (if there are), or at least details more the in vivo results (also in a dedicated paragraph).

Response: Many thanks for this comment; the application of GBMs in nerve regeneration is a newly emerging field and most of the conducted studies are at in vitro stage, and few research have moved to preclinical animal models. More details and new case studies (research articles) have been added for in vivo results.

5- I understand from 2.5 that there are not clinical studies for graphene-enhanced scaffolds for PNS, is it correct?

Response: Yes, so far, the use of GBMs for nerves is in preclinical animal studies and may translate to the clinical setting within the next few years since these materials have represented promising results.

6- Line15-20 page 16: this part should be in a par. About preclinical or clinical studies, not ion the conclusions. Maybe in the conclusions the authors should add their view of the next 3-5-10 years about graphene-based scaffold future.

Response: It has been edited based on the comment.

Reviewer 2 Report

The review manuscript " Graphene-based materials prove to be a promising candidate for nerve regeneration following peripheral nerve injury" gives an overview where graphene, graphene oxide and rGO as well others are applied in nerve cell regeneration.

There some issues with abbreviations so the reviewer suggest to make on beginning of manuscript a Table define all those applied in the manuscript because its very difficult to read the text to switch back what all abbreviation means. As well some abbreviation not really needed its better to write the meaning which makes the manuscript better understandable.

There need to be made a better introduction (visible with a scheme) where the function of Schwann cells are addressed and how graphene compounds can repair those. (Please make it simple ). A flow chart like in the middle the graphene compounds and the material applied for such as you shown with scaffolds hydrogels ets will give a better overview for readers what to expect from this review. Please add those. 

The most important issue are the graphene compound which should be the first section in view of their structure, their toxicity regarding cells etc. A small part of such is shown in 2 sentences on end of manuscript but the effects of graphene in vivo need to be shown especially where the limitations (being magnetic, nanoparticles, having negative effect if enter blood stream as well certain organs as heart, lung etc). Please provide a better introduction of the main compounds of graphene applied in this review. 

The authors also shown in several sections as well Table 2 conductivity as well modulus of such scaffolds where graphene compounds applied. Why are such relevant in medical term of nerve generation?  Which clinical studies made of such compounds? How the letal percentage in mice or other lab animals if such are made? 

The other issue if using nanoparticles of graphene compounds how do such particles dissolve in vivo or do they stay in the body? In most cases for tissue engineering those materials applied need to be biocompatible but also degradable in vivo. How does graphene compound fulfill such? This are important question that need to be addressed in the review.

Author Response

The review manuscript " Graphene-based materials prove to be a promising candidate for nerve regeneration following peripheral nerve injury" gives an overview where graphene, graphene oxide and rGO as well others are applied in nerve cell regeneration.

1- There some issues with abbreviations so the reviewer suggest to make on beginning of manuscript a Table define all those applied in the manuscript because its very difficult to read the text to switch back what all abbreviation means. As well some abbreviation not really needed its better to write the meaning which makes the manuscript better understandable.

Response: Many thanks for pointing this out. The abbreviation section has been added.

2- There need to be made a better introduction (visible with a scheme) where the function of Schwann cells are addressed and how graphene compounds can repair those. (Please make it simple). A flow chart like in the middle the graphene compounds and the material applied for such as you shown with scaffolds hydrogels ets will give a better overview for readers what to expect from this review. Please add those.

Response: Thanks for the comment; more details related to Schwann cells have been added in the introduction. Also, a new figure, Figure 2, has been added. The effect(s) of GBMs on Schwann cells is discussed case by case within the manuscript. Furthermore, our manuscript has graphical abstract which had illustrated before and it is completely aligned with your suggestion. Below is our graphical abstract illustration. 

3- The most important issue are the graphene compound which should be the first section in view of their structure, their toxicity regarding cells etc. A small part of such is shown in 2 sentences on end of manuscript but the effects of graphene in vivo need to be shown especially where the limitations (being magnetic, nanoparticles, having negative effect if enter blood stream as well certain organs as heart, lung etc). Please provide a better introduction of the main compounds of graphene applied in this review. 

Response: Thank you for the valuable comment. As mentioned within the manuscript, “GBMs can enter the body with different routes and penetrate through physiological barriers like a blood-air barrier, blood-testis barrier, blood-brain barrier (BBB) and blood-placental barrier; after that, they can locate in cells, tissues, and organs, and finally being excreted. This entrance can result in toxicity and genotoxicity, and various factors affect it, including the lateral size, surface structure, functionalisation, charge, impurities, and aggregations. Among several mechanisms underlying GBMs toxicity, the production of reactive oxygen species (ROS) within cells can lead to interactions with biomolecules (e.g., DNA and RNA) [15,19]. However, GBMs have flexible structures that can be modified with other substances, including polymers and biomolecules, to enhance their biocompatibility. GBMs composites have represented appropriate interaction with DNA and RNA for sensing and drug delivery approaches [2,20].” In this review paper, we attempted to cover important discussed aspect of each study, but unfortunately in nerve regeneration papers toxicity aspects did not discussed in depth, infact it is a challenge in this area and this is the reason that we discussed in the “Associated Challenges of GBMs in Clinical Studies” section of our manuscript. Thus, you are completely right, the toxicity of the resultant scaffolds regarding cells is one the most important issue that should be addressed in studies. Based on our research, recent studies, which are focused on the regeneration of the toxicity sensitive tissues such as lung, liver, intestine, kidney, and heart have discussed the toxicity aspect extensively, which is not our focus in this manuscript. Due to the suggestion, we have searched again and we found just a recently published article in nerve regeneration that investigate the toxicity of the fabricated scaffold in the internal organs (liver, kidney, lung, heart, and spleen) in the long term repair. We added the results of this article in the revised manuscript. Also, we added some further explanations in the introduction section.

Thank you so much for the comment. Since this manuscript is aligned with our previous review paper “Graphene Oxide: Opportunities and Challenges in Biomedicine”, we wrote this manuscript more concise and more focused on the nerve studies. However, we added some parts related to the G compounds in the introduction, based on your concern. 

4- The authors also shown in several sections as well Table 2 conductivity as well modulus of such scaffolds where graphene compounds applied. Why are such relevant in medical term of nerve generation? Which clinical studies made of such compounds? How the letal percentage in mice or other lab animals if such are made? 

Response: There is no clinical study yet regarding the application of GBMs in nerve regeneration. Thus, in this review paper, we attempted to give insight to emerging research by comparing important parameters of applying GBMs in nerve regeneration. The most important parameters, as mentioned, consist of G concentration, electrical conductivity, and mechanical properties. The more similarity of the mechanical characteristic of the fabricated scaffold to the native tissue results the more successful in the in vitro and in vivo studies. Subsequently, it will result that the fabricated scaffold utilizes in the clinical studies. Except the importance of mechanical characteristic, the electrical conductivity of the resultant scaffold plays an important role in nerve regeneration, which is an excitable tissue. We know that electrical signals and external stimuli enhance tissue regeneration for excitable tissue, specifically nerve tissue. Depolarization and repolarization take place between the two sides of nerve cell membrane under an action potential, which leads to their contracting activity and response to electrical signals. Conductive materials, such as GOBMs, with similar electrical conductivity to native tissue, are promising scaffolds that stimulate cell proliferation and differentiation in stem-cell-based therapy, with decreased cytotoxicity and improved mechanical properties.

The question regarding the lethal percentage of G: 

The applied concentration of G could be varied based on its application, for instance, the applied concentration in the biosensor is different from tissue engineering; thus, there is no mentioned specific lethal concentration in articles. Furthermore, studies preferred to apply a lower concentration of G (same as most previous studies) and instead they analyse other parameters such as varied ES intensity. This might be the reason that there is no determined concentration. In general, in vivo studies have applied the concentration of below 10%.

5- The other issue if using nanoparticles of graphene compounds how do such particles dissolve in vivo or do they stay in the body? In most cases for tissue engineering those materials applied need to be biocompatible but also degradable in vivo. How does graphene compound fulfil such? This are important question that need to be addressed in the review.

Response: Thanks for the precise comment. GBMs are not degradable by themselves, which is why they are accompanied by other polymers or accelerating agents. Other polymers stimulate the degradation of GBMs to prevent staying in the body. In general, we know that presence of G-based nanomaterials for a long time in the body may be a big challenge that should address. Based on our research, all available studies in nerve regeneration are in the pre-clinical stage, and even they do not have precise in vivo analyses for cytotoxicity or genotoxicity. Thus, the most crucial parameter that all studies have focused is GBMs concentration, and we did our best to discuss GBMs concentration in different composite scaffolds in our manuscript.

Round 2

Reviewer 2 Report

The manuscript in much better form and has enhanced clarity. All open question sufficient answered. I suggest accept as it is.

Author Response

Would like to thanks the reviewers and his positive comments.